# CLIC1 down-regulates Nrf2/HO-1 signalling pathway promoting the apoptosis and pyroptosis in OGD/R-treated HT22 cells

Jingtong Xiong[1,2☯], Shuo Li[1☯], Honghai Chen[1], Chuanwen Yu[1], Yuhai Cao[1], Xiaofeng Qu[1], Jianlin Wu[2,3], Aodan Zhang[1]*, Chuang Sun[1]*

**1** Department of Radiology, The Second Hospital of Dalian Medical University, Dalian, China, **2** Graduate School, Tianjin Medical University, Tianjin, China, **3** Department of Radiology, Affiliated Zhongshan Hospital of Dalian University, Dalian, China

☯ These authors are contributed equally to this work.
* drsunch@163.com (CS); 370151191@qq.com (AZ)

## Abstract

Cerebral ischemia–reperfusion injury (CIRI) occurs during the treatment of ischemic stroke when the affected blood vessels are recanalized and the oxygen supply to the brain is restored. Chloride intracellular channel 1 (CLIC1) and the nuclear factor erythroid-related factor 2 (Nrf2)/heme oxygenase-1 (HO-1) pathway have been implicated in many neurological disorders. However, the exact mechanism by which CLIC1 contributes to CIRI remains unclear, and its potential role in modulating the Nrf2/HO-1 signaling pathway in CIRI has yet to be explored. We investigated the potential roles of CLIC1 in CIRI using an oxygen and glucose deprivation/reoxygenation (OGD/R) model in HT22 cells. The findings of our study indicated that CLIC1 was high-expressed after OGD/R and had an inhibitory effect on the Nrf2/HO-1 signalling pathway. This process led to an exacerbation of apoptosis due to oxidative stress and an increase in the activity of the nucleotide-binding oligomeriation domain-like receptor protein 3 (NLRP3) inflammasome and pyroptosis in OGD/R-treated HT22 cells. These effects were reversed when CLIC1 was silenced. Together, the results of this study confirm our hypothesis that CLIC1 promotes CIRI by suppressing the Nrf2/HO-1 signalling pathway. CLIC1 emerges as a promising therapeutic target to prevent neuronal cell injury in CIRI.

## Introduction

Cerebral ischemia–reperfusion injury (CIRI) is a biological cascade process that exacerbates brain damage and dysfunction [1]. CIRI occurs during the treatment of ischemic stroke when the blood vessels responsible are recanalized and the oxygen supply to the brain is restored. The no-reflow phenomenon results in persistent hypoperfusion in certain areas of the already damaged brain, while

**Data availability statement:** All relevant data are within the manuscript and its Supporting Information files.

**Funding:** This research was supported by the Liaoning Provincial Science and Technology Program (2022-MS-329), the General Program of National Natural Science Foundation of China (82071911), and the Science and Technology Innovation Fund of Dalian (2021JJ12SN38). The funders had no role in study design, data collection and analysis, decision to publish, or preparation of the manuscript.

**Competing interests:** The authors have declared that no competing interests exist.

other regions, where microvascular blood flow is restored, become vulnerable to ischemia-reperfusion injury [2]. The pathophysiological mechanism of CIRI is complex, mainly involving mitochondrial dysfunction, increased reactive oxygen species (ROS) production, and neutrophil infiltration, promoting oxidative stress and inflammation [3,4]. Excessive ROS can oxidise lipids, proteins, and DNA, leading to oxidative injury and trigger apoptosis, which involves cell shrinkage, DNA fragmentation, and the formation of apoptotic bodies [5,6]. Caspases, activated by intrinsic or extrinsic cues of oxidative stress, initiate this non-inflammatory process [6]. Inflammatory responses also play a key role in CIRI [7]. Pyroptosis is a form of inflammation-mediated cell death, which results from the changes of membrane permeability, cell swelling, membrane rupture, and the release of pro-inflammatory cytokines [8]. In the canonical model of caspase-1-mediated pyroptosis, the recognition of inflammatory ligands triggers the activation of intracellular multiprotein signalling complexes called inflammasomes, while non-canonical inflammasomes have been shown to activate caspase-4/5/11 [9]. The nucleotide-binding oligomerization domain-like receptor protein 3 (NLRP3) inflammasome activates caspase-1, promoting interleukin (IL)-1β and IL-18 secretion for immune defence [10]. Furthermore, caspase-1/4/5/11 targets gasdermin D (GSDMD), a member of the gasdermin family that drives programmed cell death and pyroptosis through its membrane pore-forming activity [6,9].

In CIRI, although distinct forms of programmed cell death, including apoptosis and pyroptosis, have different mechanisms and functions, they share some common features. Cell death is not always a consequence of inflammation, but sometimes acts as a trigger [11]. Some key initiator caspases in the apoptotic pathway also regulate to pyroptosis. For example, the apoptotic effector caspase-3 can cleave GSDME to trigger pyroptosis [9]. Currently, reperfusion therapies, such as intravenous thrombolysis and mechanical thrombectomy, are limited by a narrow therapeutic window, limited applicability, and high costs, which prevent these methods from benefiting the majority of stroke patients [12–14]. Meanwhile, these invasive therapies pose a biphasic role. Rapid reperfusion by these operations is beneficial during the acute phase by restoring blood flow and oxygen supply, but it can become detrimental during the recovery phase, potentially leading to and exacerbating reperfusion injury [14]. Therefore, there is an urgent need for novel therapeutic approaches. Developing drugs targeting both apoptotic and pyroptotic pathways could offer a transformative strategy for mitigating neuronal damage.

Chloride intracellular channel 1 (CLIC1) is a multifunctional protein expressed in various organs, where it participates in redox regulation, immune and inflammatory processes via enzymatic activity [15,16]. It is an oxidative stress sensor in vascular endothelium cells [17]. In response to increased cytoplasmic oxidation or changes in pH, cytoplasmic soluble CLIC proteins integrate into cellular membranes, functioning as anion channels to counteract cationic fluctuations [18]. It is highly expressed in neurological disorders, and is indicative of a poor prognosis [19]. However, the precise mechanism of how CLIC1 promotes CIRI and the relative pathway remains largely unknown. A recent study [17] discovered that in umbilical vein endothelial cells, upregulation of CLIC1 impaired the ability of vascular cells to resist oxidative

stress to speed up cellular senescence; in contrast, CLIC1 inhibition resulted in a protective effect, preventing cellular aging and disorder. This protection function was considered to be achieved via activation of the nuclear factor erythroid-related factor 2 (Nrf2)/heme oxygenase-1(HO-1) pathway. Nrf2 bind to antioxidant response element (ARE), a specific DNA sequence located in the promoter regions of genes that encode antioxidant and cytoprotective proteins, in response to oxidative or electrophilic stress, activating the expression of these genes and enhancing the cell's defense mechanisms against damage [20]. Nrf2/HO-1 pathway is a key protector against oxidative stress-mediated apoptosis and related to inflammasome regulation and pyroptosis [21–23].

Therefore, this study presents a novel insight into the regulatory role of CLIC1 in CIRI via Nrf2/HO-1 pathway using a cell model, emphasizing its critical involvement in oxidative stress and inflammation. We hypothesized that CLIC1 is highly expressed in OGD/R-treated HT22 cells and inhibits the Nrf2/HO-1 signalling pathway. These processes promote apoptosis and NLRP3-mediated pyroptosis. Fig 1 shows a diagram of the scientific hypothesis. These findings could unveil promising targets for the treatment of stroke and CIRI, ultimately improving outcomes and broadening the therapeutic options.

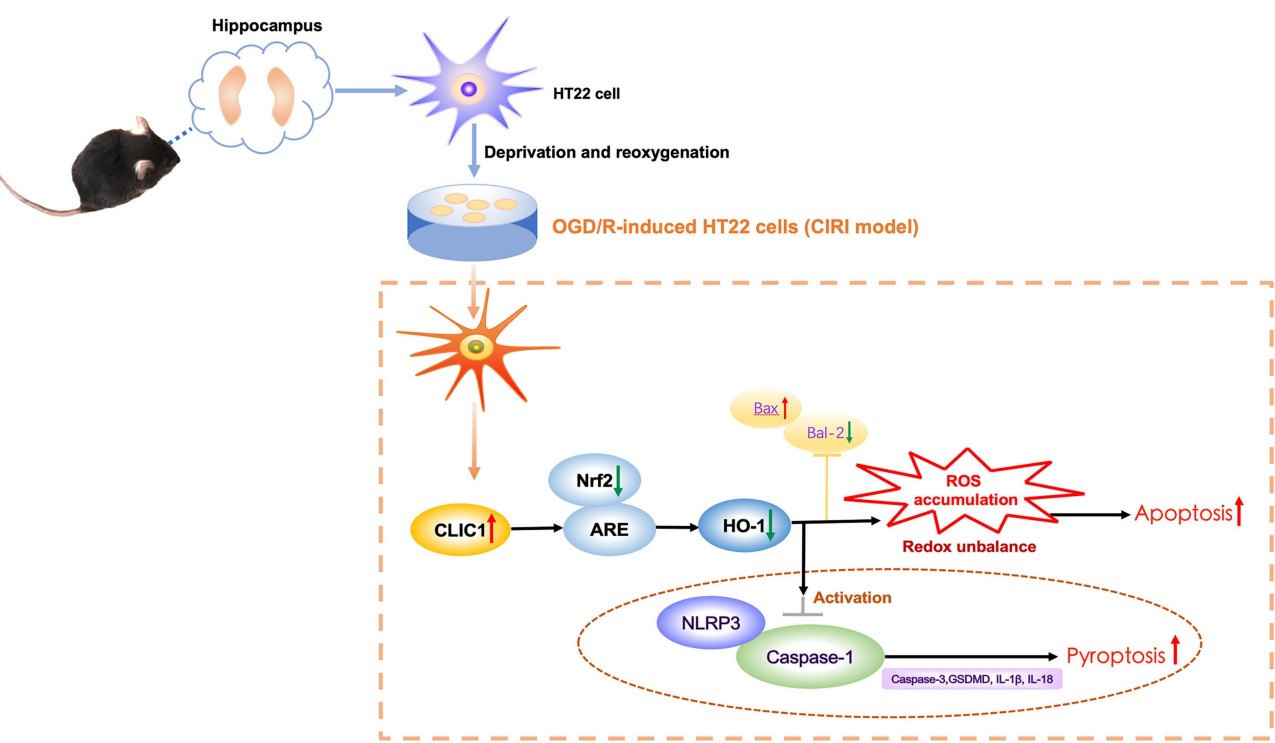

**Fig 1. Schematic illustration of the hypothesis of CLIC1 mediates neuronal injury following oxygen and glucose deprivation/reoxygenation (OGD/R) in HT22 cells.** Mouse hippocampal neurons (HT22 cells) are subjected to OGD/R to mimic CIRI. OGD/R triggers upregulation of CLIC1, which leads to suppression of the Nrf2/HO-1 antioxidant signaling pathway. This results in increased ROS accumulation and disturbance of redox balance, promoting apoptosis by increased Bax and decreased Bcl-2 expression. Additionally, elevated CLIC1 activates the NLRP3 inflammasome and caspase-1 via inhibiting the Nrf2/HO-1 pathway, inducing pyroptosis via upregulation of pyroptosis markers, such as caspase-3, GSDMD, IL-1β, and IL-18. Collectively, these processes contribute to neuronal cell damage after CIRI. ARE, antioxidant response element; CIRI, cerebral ischemia-reperfusion injury; CLIC1, chloride intracellular channel 1; GSDMD, gasdermin D; HO-1, heme oxygenase 1; IL, interleukin; NLRP3, NOD-like receptor family pyrin domain containing 3; Nrf2, nuclear factor erythroid 2–related factor 2; OGD/R, oxygen-glucose deprivation/reoxygenation; ROS, reactive oxygen species.

## Materials and methods

### Cell culture

The mouse HT22 hippocampus neuronal cell line (HT22, iCell, China) was acquired and cultivated in Dulbecco's Modified Eagle Medium (DMEM, Servicebio, China) with 10% foetal bovine serum (FBS, Biobase, China) at 37°C in a 5% CO2 incubator. This study was carried out in strict accordance with the recommendations in the guideline for the ethical review of animal welfare of laboratory animals. The protocol was approved by the Committee on the Ethics of Animal Experiments of Dalian Medical University (approval No. AEE22014).

### Model establishment and group treatments

The CIRI model was established in HT22 cells through the process of oxygen and glucose deprivation/reoxygenation (OGD/R). This study included the following groups: control, OGD/R, negative control (NC), CLIC1 silencing, OGD/R + NC, OGD/R+CLIC1 silencing, OGD/R+CLIC1 silencing+Nrf2/HO-1 inhibition, and OGD/R+CLIC1 silencing+NLRP3 activation. The HT22 cells in the control group were inoculated in 6-well culture plates and incubated overnight for cell processing. The cells were incubated in glucose-free Hank's Balanced Salt Solution (HBSS) under 95% air and 5% CO2 at 37 °C for 6h. To induce OGD/R, HT22 cells were seeded in 6-well culture plates and cultured overnight to ensure proper adherence. Hypoxia was induced by incubating the cells in glucose-free HBSS in an anoxic chamber maintained at 37°C under 95% $N_2$ and 5% $CO_2$ for 6 hours. Then in reperfusion phase following hypoxic treatment, the glucose-free HBSS was replaced with normal DMEM medium (supplemented with glucose and 10% FBS), and the cells were cultured under air conditions (37°C, 21% $O_2$, and 5% $CO_2$) for 24 hours. Cells were subsequently used for further experiments.

The control HT22 cells and OGD/R-induced cells were then infected with either the CLIC1 silencing-negative control virus (WL117218−2, Wanleibio, China) or CLIC1 silencing virus (WL117218−1, Wanleibio, China) to create the NC and CLIC1 silencing groups. The viral stock was thawed at 4°C. The virus solution was prepared by accurately measuring the required volume to achieve a multiplicity of infection (MOI) of 50 and mixing it with the culture medium. The existing cell medium was then replaced with the virus-containing medium. The cells were gently mixed to ensure even distribution and incubated at 37°C with 5% $CO_2$. After 48h, the cells were transferred to glucose-free HBSS and underwent the OGD/R protocol described above. After the CLIC1 silencing virus infection, further experiments were performed 48h later. The OGD/R+CLIC1 silencing+Nrf2/HO-1 inhibition and OGD/R+CLIC1 silencing+NLRP3 activation groups were pretreated with SnPPIX (50 μM, Macklin, China, for 3h) or Nigericin (10 μg/mL, MCE, China, for 2h), respectively. Following pretreatment, the cells were incubated in glucose-free HBSS under hypoxic conditions (37°C, 95% $N_2$, and 5% $CO_2$) for 4h. Subsequently, the cells were returned to normal DMEM medium and incubated under air conditions (37°C, 21% $O_2$, and 5% $CO_2$) for 24h before further experiments.

### Real-time polymerase chain reaction (PCR)

Gene expression levels were quantified using real-time PCR analysis. TRIpure lysate (BioTeke, Beijing, China) was used to extract total RNA from the HT22 cells. The RNA purity in each sample was assessed using a NANO 2000 UV spectrophotometer (Thermo, USA). The RNA samples were subjected to reverse transcription using BeyoRT II M-MLV reverse transcriptase (Beyotime, Shanghai, China) to obtain complementary DNA (cDNA). The SYBR Green PCR MasterMix (Solarbio, Beijing, China) was used to perform real-time PCR, following the instructions provided by the manufacturer. Quantitative analysis of mRNA levels was performed by normalizing them using a threshold cycle (Ct) of $2^{-\Delta\Delta Ct}$. β-actin was used as an internal control. Table 1 shows the forward and reverse primers.

### Western blot (WB)

Following protein extraction and subsequent separation using protein electrophoresis, the protein samples were deposited onto polyvinylidene fluoride (PVDF) membranes (IPVH00010, Millipore, USA). The membranes were immersed

**Table 1. Primer Sequences Applied in Real-Time PCR.**

| Primer | Forward 5′-3′ | Reverse 5′-3′ |
|---|---|---|
| CLIC1 | TGCCGTTCTTGCTCTATG | GGTGTTGGACTCAGGGTT |
| Nrf2 | GTGCTCCTATGCGTGAA | GCGGCTTGAATGTTTGT |
| HO-1 | ACAGATGGCGTCACTTCG | TGAGGACCCACTGGAGGA |
| Bcl-2 | TGTGCCACCTGTGGTCCATC | CATCTCCCTGTTGACGCTCT |
| Bax | GCTACAGGGTTTCATCCAGG | GTCCACGTCAGCAATCATCC |
| NLRP3 | GAGTTCTTCGCTGCTATGT | ACCTTCACGTCTCGGTTC |
| Caspase-1 | GGGACCACATACTCTAAT | TAATGCCATCATCTTCA |
| Caspase-3 | TGGGACTGATGAGGAGA | ACTGGATGAACCACGAC |
| GSDMD | GCTTTATGCTTGAAGGGTG | AAGGTCCTCTGTTTCTCATCT |
| IL-1β | CTCAACTGTGAAATGCCACC | GAGTGATACTGCCTGCCTGA |
| IL-18 | GGCTGCCATGTCAGAAGA | CCGTATTACTGCGGTTGT |
| β-actin | AATCGTGCGTGACATCAA | AGAAGGAAGGCTGGAAAA |

in a blocking solution and then treated with primary antibodies targeting CLIC1 (1:1000, 14545–1-AP, Proteintech, China), Nrf2 (1:500, WL02135, Wanleibio, China), HO-1 (1:500, WL02400, Wanleibio, China), B-cell lymphoma-2 (Bcl-2: 1:400, WL01556, Wanleibio, China), Bax (1:1000, WL01637, Wanleibio, China), caspase-3/cleaved caspase-3 (1:400, WL02117, Wanleibio, China), NLRP3 (1:500, WL02635, Wanleibio, China), pro caspase-1/ cleaved caspase-1 (1:500, WL03450,Wanleibio,China), IL-1β (1:1000, WL00891, Wanleibio, China), IL-18 (1:500, WL01127, Wanleibio, China), or GSDMD (1:500, AF4012, Affinity, China) overnight at 4°C. After undergoing several washes with Tris-buffered saline containing Tween (TBST), the membranes were then exposed to the secondary antibody of (goat anti-rabbit IgG-HRP, 1:5000, WLA023, Wanleibio, China), and incubated at 37°C for 45 min. The anti-rabbit β-actin antibody (1:1000, WL01372, Wanleibio, China) and histone H3 antibody (1:1000, WL0984a, Wanleibio, China) were used as reference antibodies for normalization. The films were scanned and the OD values of the target bands were assessed using Gel-Pro analyser software.

## Flow cytometry

Flow cytometric analyses were used to quantify the rate of cell apoptosis and the quantity of ROS. Concisely, the cells were gathered and washed twice with phosphate-buffered saline (PBS). Then, the cells were treated with Annexin V-FITC and propidium iodide for 15 minutes under dark conditions at ambient temperature in order to evaluate apoptosis. Next, the cells were exposed to a 1:1000 dilution of DCFH-DA and kept in an incubator at a temperature of 37°C for 20 minutes. Ultimately, the samples underwent three rounds of washing with PBS prior to evaluating the ROS level. The experiment was performed following the manufacturer's instructions provided by the cell apoptosis kit (WLA001, Wanleibio, China) or the reactive oxygen detection kit (WLA131, Wanleibio, China).

## Assessment of IL-1β and IL-18 levels, GSH-Px, SOD, and CAT activity, and LDH leakage rate

The gene and protein expression levels of IL-1β and IL-18 were quantified using real-time PCR and WB, as described above. Additionally, the protein levels in solution were also detected using enzyme-linked immunosorbent assay (ELISA) kits, namely IL-1β (WLE03, Wanleibio, China) and IL-18 (EK218, Liankebio, China). And reagent kits were used to measure the activity of glutathione peroxidase (GSH-Px: WLA107, Wanleibio, China), superoxide dismutase (SOD: WLA110, Wanleibio, China), catalase (CAT: A007, Jianchengbio, China), and leakage rate of lactate dehydrogenase (LDH: WLA072, Wanleibio, China). All the assays were conducted in accordance with the manufacturer's instructions.

### Cell Counting Kit-8 (CCK-8)

HT22 cells were seeded into a 96-well plate with five replicates per group and incubated in standard conditions overnight. Subsequently, the cells were infected with a virus and underwent processing 48 hours later. The CCK-8 solution (10 μl, WLA074, Wanleibio, China) was added to each well for incubation for 1 h at 37°C in 5% $CO_2$. Data analysis was performed on an enzyme marker (800Ts, BIOTEK, USA) by measuring the optical density (OD) value at 450 nm.

### Statistical analysis

SPSS 26.0 software (IBM Corporation, NY, USA) was used for statistical analyses. Comparisons between the two groups were conducted using a two-tailed *t* test. Multiple group comparisons were conducted using a one-way analysis of variance (ANOVA). For multiple comparisons, the Bonferroni test was applied. Spearman correlation analysis was conducted among the expression levels of genes detected by real-time PCR. *P* < 0.05 was considered to be statistically significance.

## Results

### CLIC1 is highly expressed in OGD/R-treated HT22 cells

We assessed the gene and protein expression of CLIC1 by real-time PCR and WB, respectively. Following exposure to OGD/R, HT22 cells exhibited higher mRNA and protein levels of CLIC1 compared with control cells (*P* = 0.014 and *P* < 0.001, respectively). After viral silencing of CLIC1 in HT22 cells, we saw a reduction in the CLIC1 expression level (Fig 2a-c).

### Upregulation of CLIC1 promoted oxidative stress-induced apoptosis in OGD/R-treated HT22 cells

Flow cytometric analysis showed that the upregulation of CLIC1 in OGD/R-exposed HT22 cells led to an increased apoptosis rate and mean fluorescence intensity (MFI) value for ROS. These levels were reduced by CLIC1 silencing (Fig 2d-f). Furthermore, increased CLIC1 expression in OGD/R-exposed HT22 cells resulted in decreased levels of SOD, CAT, and GSH-Px activity as well as a lower CCK-8 OD value and a higher LDH level. These changes were reversed by silencing CLIC1 (Fig 3a-e). Additionally, the expression level of Bcl-2 was lower and that of Bax was higher after OGD/R (Fig 3f-h). Upon CLIC1 silencing, both of the two indicators exhibited an inverse trend. These results indicate that increased expression of CLIC1 results in a redox imbalance in OGD/R-treated HT22 cells, which then aggravates oxidative stress-induced apoptosis.

### CLIC1 upregulation promoted NLRP3-mediated pyroptosis in OGD/R-treated HT22 cells

OGD/R resulted in the elevation of CLIC1, leading to higher expression levels of NLRP3, caspase-1, caspase-3, GSDMD, IL-1β, and IL-18, according to real-time PCR, WB, and/or ELISA analyses. Conversely, when CLIC1 was silenced, the expression levels of the aforementioned markers were all reduced (Fig 4). We further activated NLRP3 after silencing CLIC1 to confirm that NLRP3-mediated pyroptosis occurs in the OGD/R-treated HT22 cells. With the activation of NLRP3, the levels of caspase-1, caspase-3, GSDMD, IL-1β, and IL-18 were all increased on real-time PCR, WB, and/or ELISA analyses in the OGD/R+CLIC1 silencing+NLRP3 activation group compared with the corresponding levels in the OGD/R+CLIC1 silencing group. These results indicate that pyroptosis can be promoted by activation of NLRP3 in OGD/R-exposed HT22 cells. Together, these findings suggest that upregulation of CLIC1 improves NLRP3-mediated pyroptosis in OGD/R-exposed HT22 cells.

### CLIC1 inhibits the Nrf2/HO-1 signalling pathway to promote apoptosis

The expression levels of Nrf2 and HO-1 in HT22 cells were reduced after OGD/R. Upon CLIC1 silencing, their expression levels were increased; however, they did not fully return to normal levels in all groups after OGD/R. We then applied CLIC1

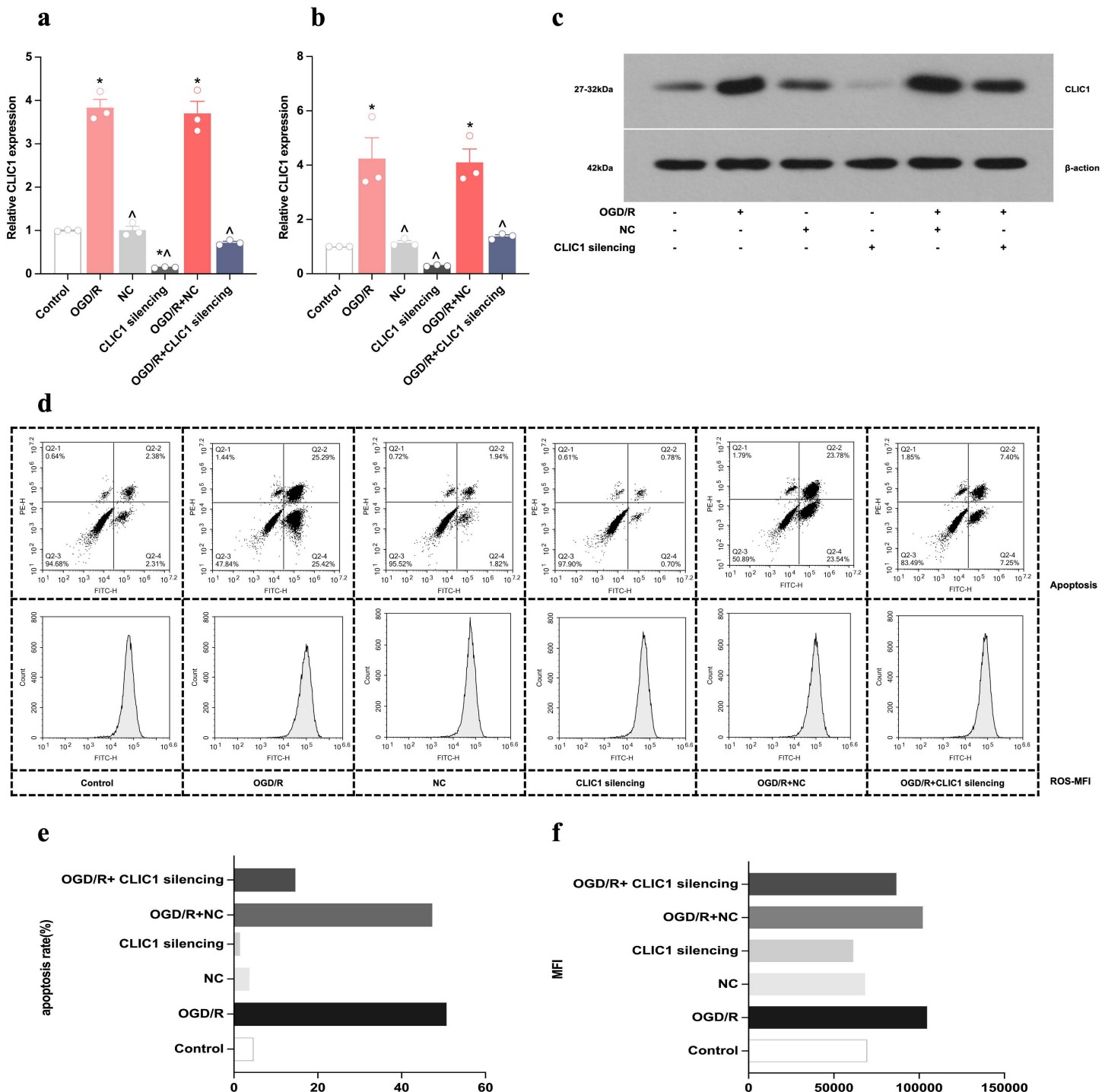

**Fig 2. CLIC1 upregulates and promotes apoptosis and reactive oxygen species (ROS) accumulation in oxygen and glucose deprivation/reoxygenation (OGD/R)-treated HT22 cells.** (a) Real-time PCR analysis of CLIC1 gene expression showed significantly higher expression in the OGD/R group compared to the control group, with a marked reduction upon CLIC1 silencing. (b, c) Western blot analysis of CLIC1 protein levels demonstrated increased CLIC1 expression in the OGD/R group and its suppression following CLIC1 silencing. (d-f) Flow cytometry analysis shows that elevated CLIC1 expression in OGD/R-treated HT22 cells led to an increased apoptosis rate (d, e) and higher ROS mean fluorescence intensity (MFI) values (d, f). These effects were reversed by silencing CLIC1. $^*P < 0.05$ versus control group; $^\wedge P < 0.05$ versus OGD/R group.

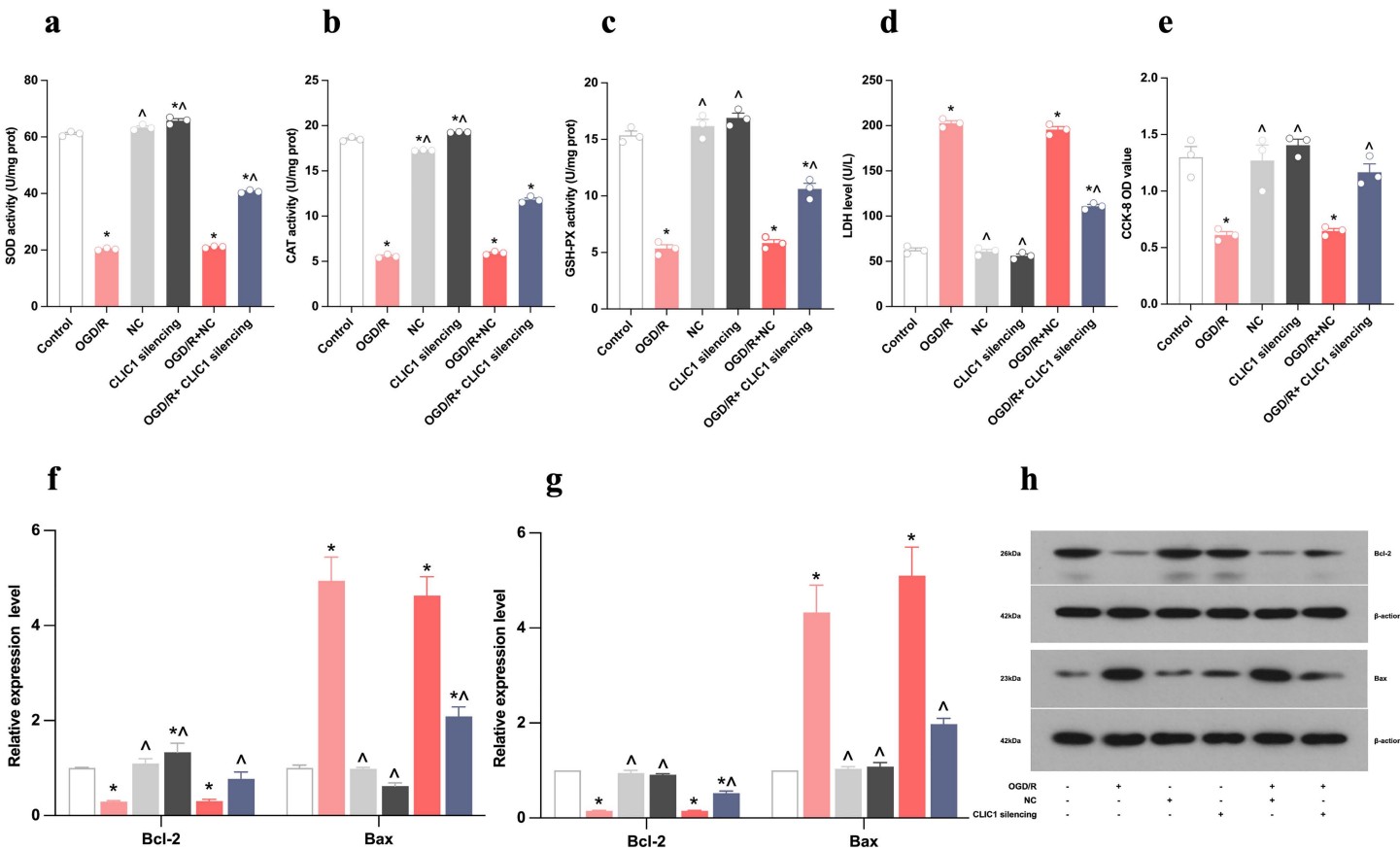

**Fig 3. CLIC1 upregulation accelerates oxidative stress in oxygen and glucose deprivation/reoxygenation (OGD/R)-treated HT22 cells.** (a-c) The activities of SOD (a), CAT (b), and GSH-Px (c) were significantly reduced in OGD/R-treated HT22 cells compared to the control group, with the restoration of these antioxidant enzyme activities upon CLIC1 silencing. (d) The LDH level was elevated in the OGD/R group and decreased following CLIC1 silencing. (e) The CCK-8 OD value was reduced in the OGD/R group and increased with CLIC1 silencing. (f-h) Real-time PCR (f) and WB (g,h) analyses showed that OGD/R treatment decreased anti-apoptotic Bcl-2 expression and increased pro-apoptotic Bax expression, with these effects reversed by CLIC1 silencing. Data are presented as mean±SEM, with $n=3$ per group. $^*P<0.05$ versus control group; $^\wedge P<0.05$ versus OGD/R group.

silencing and inhibition of Nrf2/HO-1 signalling to explore whether the latter is the pathway responsible for the apoptosis in OGD/R HT22 cells. After inhibition of the Nrf2/HO-1 pathway, Nrf2 and HO-1 expression levels were obviously lower in the OGD/R+CLIC1 silencing+Nrf2/HO-1 inhibition group than in the OGD/R+CLIC1 silencing group (Fig 5a-c). Subsequently, the apoptosis rate and ROS MFI value were both higher in the OGD/R+ CLIC1 silencing+ Nrf2/HO-1 inhibition group compared with those in the OGD/R+CLIC1 silencing group (Fig 6a-c). In addition, the levels of SOD, CAT, GSH-Px activity and CCK-8 OD value were all reduced while the LDH level was increased after silencing CLIC1 and inhibiting Nrf2/HO-1 expression, and these changes were greater than those observed after merely silencing CLIC1 after OGD/R (Fig 7a-e). Furthermore, inhibition of Nrf2/HO-1 signalling resulted in a decrease in Bcl-2 expression and an increase in Bax expression (Fig 7f-h). These findings indicate that the Nrf2/HO-1 signalling pathway is important in neuronal protection, which alleviates oxidative stress-induced apoptosis. These processes are regulated by CLIC1 expression in OGD/R-induced HT22 cells.

## CLIC1 inhibits the Nrf2/HO-1 signalling pathway to accelerate pyroptosis

As explained above, the increased CLIC1 led to decreased Nrf2/HO-1 expression. We also found that the expression levels of inflammation- and pyroptosis-related indicators NLRP3, caspase-1, caspase-3, GSDMD, IL-1β, and IL-18 were

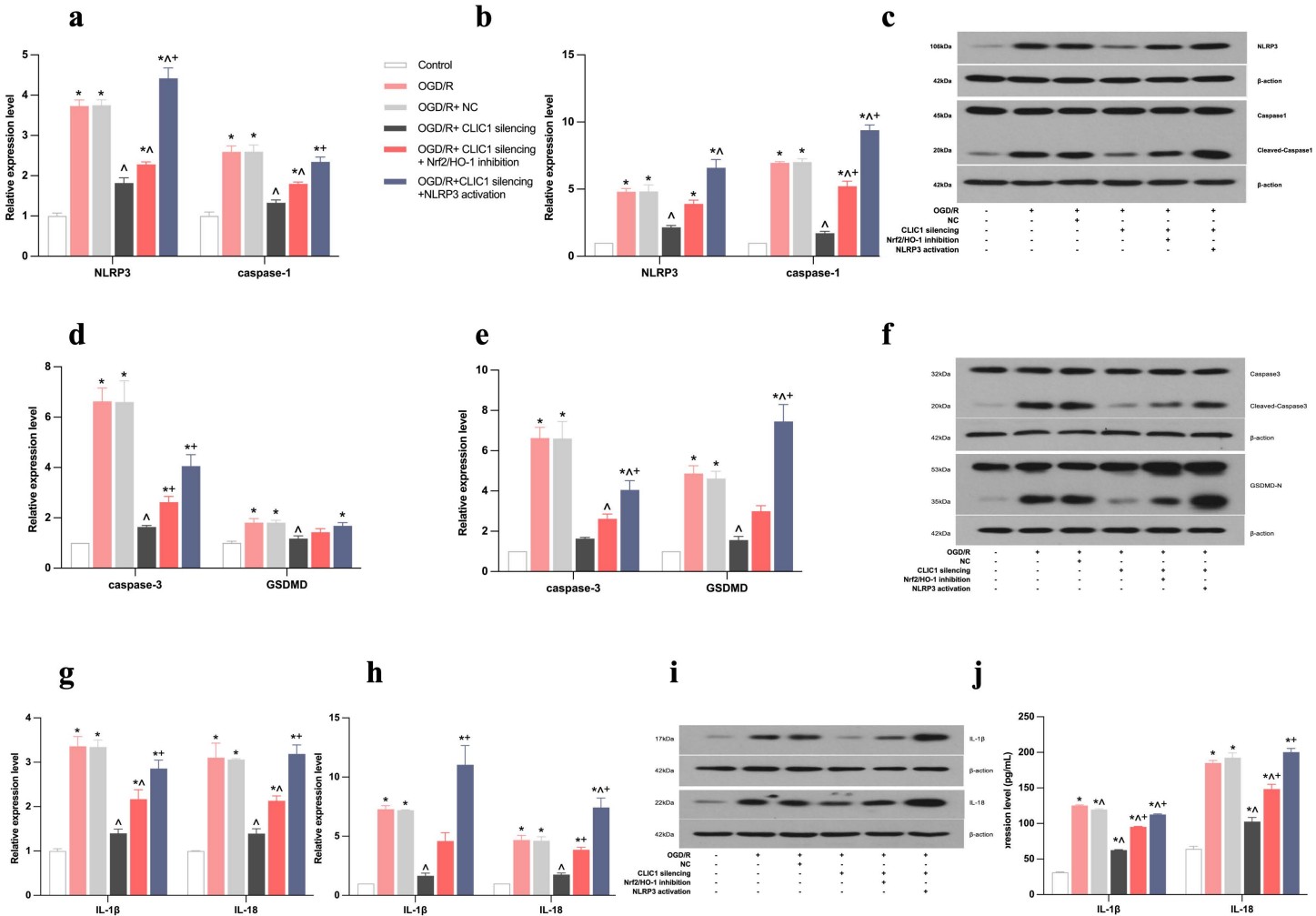

**Fig 4. CLIC1 promotes NLRP3-mediated pyroptosis by inhibiting the Nrf2/HO-1 signalling pathway in oxygen and glucose deprivation/reoxygenation (OGD/R)-treated HT22 cells.** Real-time PCR, WB, and/or ELISA analyses show that the expression levels of inflammation- and pyroptosis-related indicators were all upregulated after the increased expression of CLIC1 in OGD/R-treated HT22 cells, via inhibiting the Nrf2/HO-1 pathway. (a-c) Real-time PCR (a) and WB (b, c) analyses showed upregulation of NLRP3 and caspase-1 expression in OGD/R-treated HT22 cells, with significant downregulation upon CLIC1 silencing. Inhibition of Nrf2/HO-1 signalling or activation of NLRP3 partly reversed the effects of CLIC1 silencing. (d-f) Real-time PCR (d) and WB (e, f) analyses of caspase-3 and GSDMD protein expression showed similar trends with NLRP3 and caspase-1. (g-j) Real-time PCR (g), WB (h,i), and ELISA (j) analyses revealed elevated levels of IL-1β and IL-18 in OGD/R-treated HT22 cells, which were reduced upon CLIC1 silencing. These cytokines were further upregulated with Nrf2/HO-1 inhibition or NLRP3 activation after CLIC1 silencing. Quantitative data were presented as mean ± SEM with three replicate experiments. *$P < 0.05$ versus control group; ^$P < 0.05$ versus OGD/R group; +$P < 0.05$ vs OGD/R+CLIC1 silencing group.

all increased after inhibiting the Nrf2/HO-1 pathway in HT22 cells treated with OGD/R, and their levels were higher in the OGD/R+CLIC1 silencing+ Nrf2/HO-1 inhibition group than in the OGD/R+CLIC1 silencing group (Fig 4). Thus, NLRP3-mediated pyroptosis could be accelerated by inhibiting the Nrf2/HO-1 signalling pathway. Conversely, comparing the OGD/R+CLIC1 silencing+ NLRP3 activation group to the OGD/R+CLIC1 silencing group, the activation of NLRP3 led to increased apoptosis rates (Fig. 6a, b) and MFI value of ROS (Fig. 6a, c). Meanwhile, NLRP3 activation was also found to exacerbate oxidative stress by downregulating the level of oxidative stress–related indices of SOD, CAT, GSH-PX, CCK-8 and Bal-2, and upregulating the level of LDH and Bax (Fig. 7). All these findings imply that NLRP3-mediated

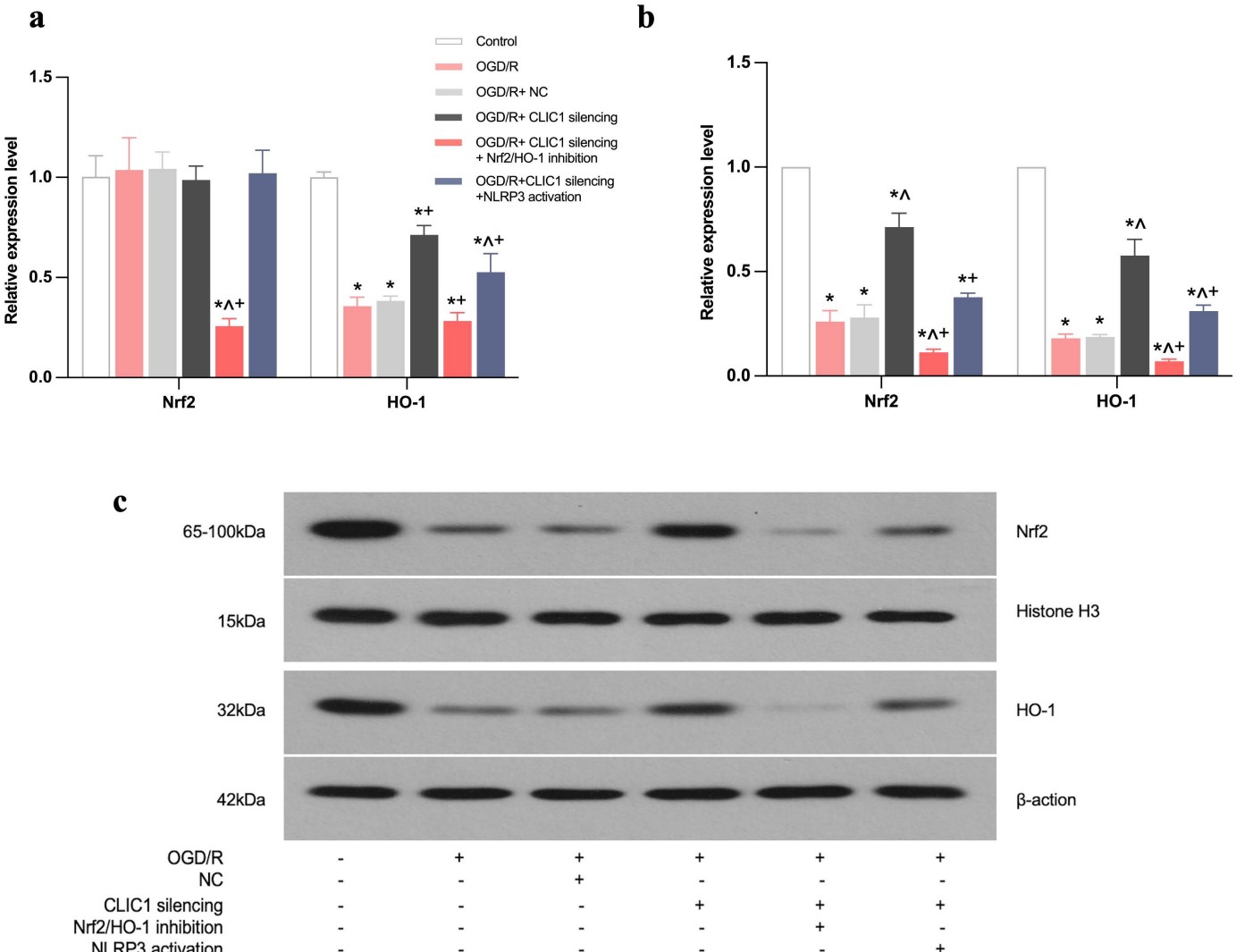

**Fig 5. High CLIC1 expression inhibits the Nrf2/HO-1 signalling pathway in oxygen and glucose deprivation/reoxygenation (OGD/R)-treated HT22 cells.** (a-c) Real-time PCR (a) and WB (b, c) analyses showed that Nrf2 and HO-1 expression levels were significantly downregulated in OGD/R-treated HT22 cells compared to the control group. CLIC1 silencing partially restored Nrf2 and HO-1 expression levels, although normal levels were not fully achieved in any group following OGD/R treatment. Data are shown as mean ± SEM, with $n = 3$ per group. $^{*}P < 0.05$ versus control group; $^{\wedge}P < 0.05$ versus OGD/R group; $^{+}P < 0.05$ vs OGD/R+CLIC1 silencing group.

pyroptosis can in turn amplify oxidative stress and apoptotic processes, indicating these events may occur in a sequential or partially overlapping manner rather than as mutually exclusive forms of cell death.

### Correlation of CLIC1 expression with apoptosis and pyroptosis-related genes

Fig 8 displays the heat maps of gene expression obtained from real-time PCR and the Spearman correlation analysis. The Spearman correlation analysis revealed a negative relationship between the gene expression level of CLIC1 and HO-1 ($r = -0.40$, $P = 0.015$) as well as Bcl-2 ($r = -0.59$, $P < 0.001$), and CLIC1 was positively related with Bax ($r = 0.63$, $P < 0.0001$). Meanwhile, the expression level of CLIC1 showed a positive correlation with NLRP3, caspase-1, caspase-3, GSDMD,

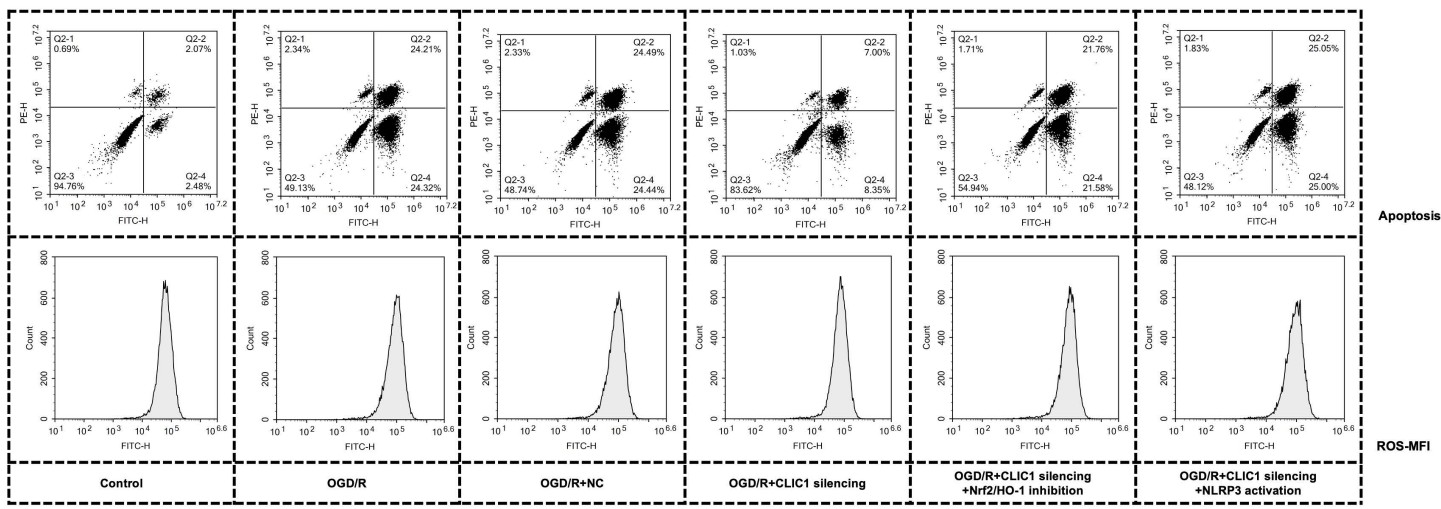

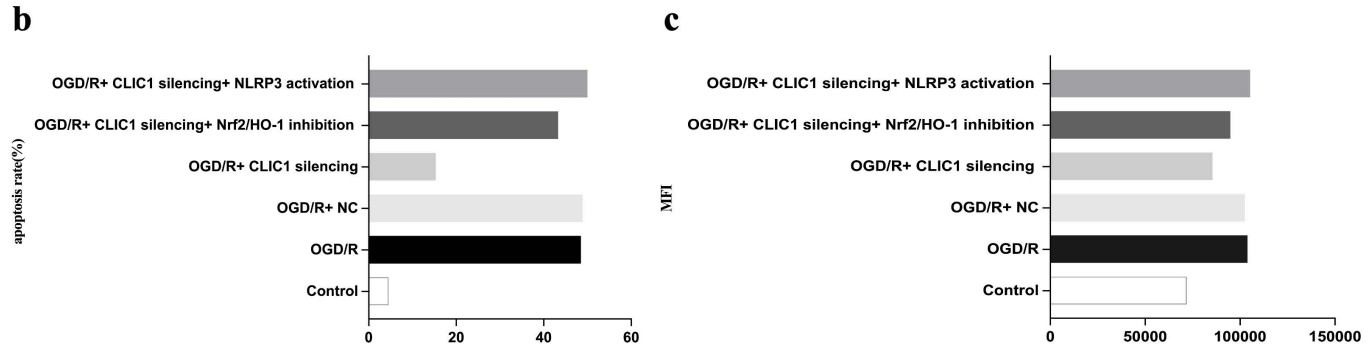

**Fig 6. CLIC1 exacerbates apoptosis and reactive oxygen species (ROS) accumulation in oxygen and glucose deprivation/reoxygenation (OGD/R)-treated HT22 cells by inhibiting the Nrf2/HO-1 signalling pathway.** Flow cytometry analysis shows that inhibition of the Nrf2/HO-1 signalling pathway increased both the apoptosis rate (a, b) and ROS mean fluorescence intensity (MFI) value (a, c) in OGD/R-treated HT22 cells. These levels were higher in the OGD/R+CLIC1 silencing+Nrf2/HO-1 inhibition group compared to the OGD/R+CLIC1 silencing group.

IL-1β, and IL-18 ($r=0.45$, $P=0.006$; $r=0.62$, $P<0.0001$; $r=0.44$, $P=0.007$; $r=0.63$, $P<0.0001$; $r=0.65$, $P<0.0001$; $r=0.53$, $P<0.001$, respectively). In combination with the results presented above, these findings further suggest that CLIC1 promotes neuronal oxidative stress-induced apoptosis and NLRP3-mediated pyroptosis by inhibiting the Nrf2/HO-1 signalling pathway in OGD/R-treated HT22 cells.

## Discussion

Our study aimed to assess the role of CLIC1 in modulating cellular responses to oxidative stress and inflammation in HT22 cells under OGD/R conditions, mimicking CIRI. CLIC1 expression was significantly elevated in OGD/R-treated HT22 cells, suggesting its role as a stress-responsive protein in cellular metabolism. Notably, following the targeted silencing of CLIC1, its expression was effectively downregulated, demonstrating the modulation achievable through genetic interventions in CIRI. CLIC1 upregulation promoted pro-apoptotic effects increasing apoptosis rate, evidenced by the increased oxidative stress markers such as ROS and LDH, decreased antioxidant enzyme activity (SOD, CAT, GSH-Px), and

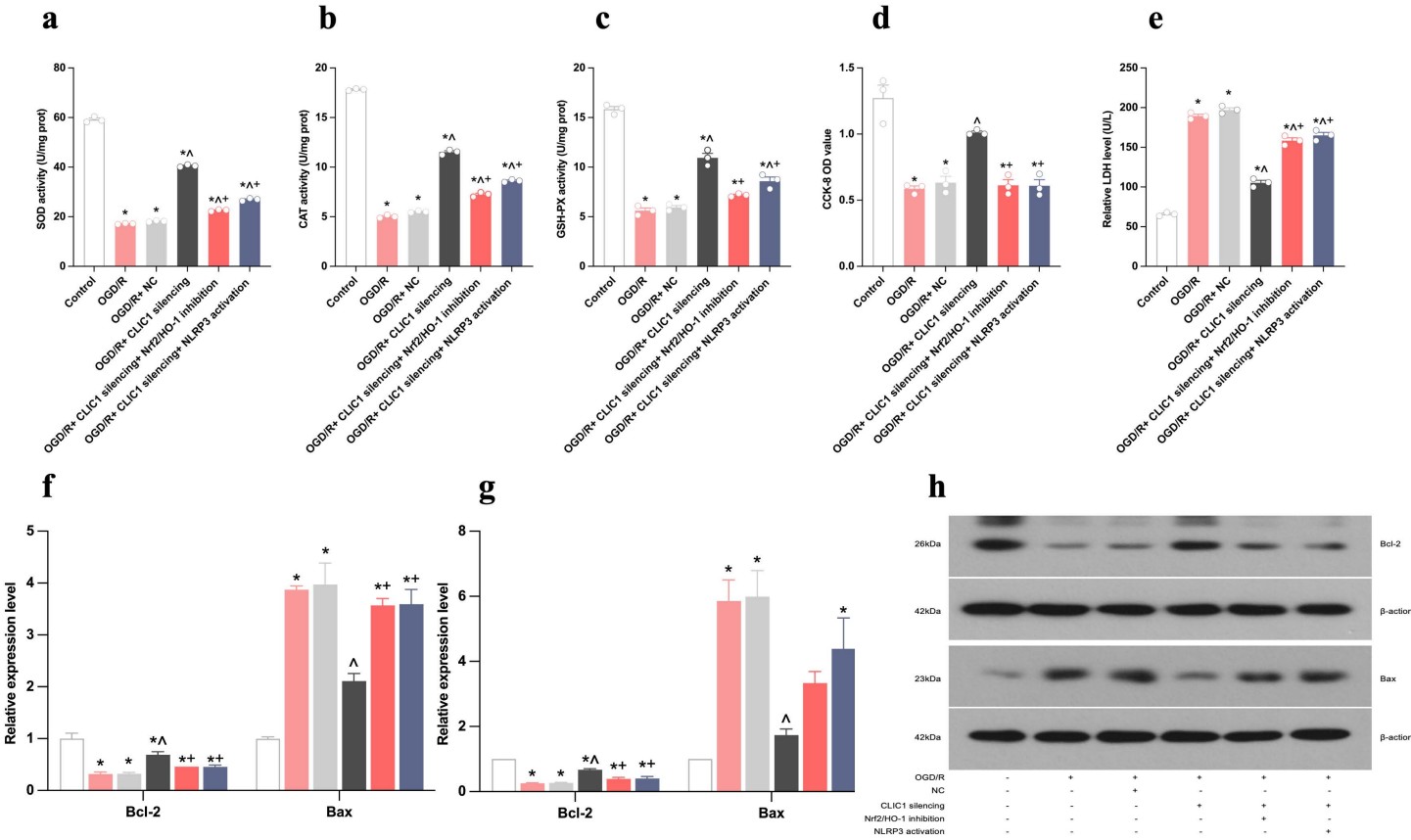

**Fig 7. CLIC1 accelerates oxidative stress in oxygen and glucose deprivation/reoxygenation (OGD/R)-treated HT22 cells by inhibiting the Nrf2/HO-1 signalling pathway.** (a-c) Activity of oxidative stress markers of SOD (a), CAT (b), and GSH-Px (c) was reduced following inhibition of the Nrf2/HO-1 signalling pathway. These reductions were more pronounced in the OGD/R+CLIC1 silencing+Nrf2/HO-1 inhibition group compared to the OGD/R+CLIC1 silencing group. (d) CCK-8 OD values were decreased after Nrf2/HO-1 inhibition, with lower values observed in the OGD/R+CLIC1 silencing+Nrf2/HO-1 inhibition group than the OGD/R+CLIC1 silencing group. (e) The LDH levels were increased after Nrf2/HO-1 inhibition and were significantly higher in the OGD/R+CLIC1 silencing+Nrf2/HO-1 inhibition group compared to the OGD/R+CLIC1 silencing group. (f-h) Real-time PCR (f) and WB (g, h) analyses showed that Bcl-2 expression was downregulated, while Bax expression was upregulated following Nrf2/HO-1 inhibition. These effects were more pronounced in the OGD/R+CLIC1 silencing+Nrf2/HO-1 inhibition group than the OGD/R+CLIC1 silencing group. Data were presented as mean±SEM, with $n=3$ per group. $^*P<0.05$ versus control group; $^\wedge P<0.05$ versus OGD/R group; $^+P<0.05$ vs OGD/R+CLIC1 silencing group.

reduced CCK-8 levels. Furthermore, our study demonstrated that CLIC1 facilitates NLRP3-mediated pyroptosis in OGD/R HT22 cells, with elevated CLIC1 correlating with increased inflammasome and pyroptosis markers, including NLRP3, caspase-1, caspase-3, GSDMD, IL-1β, and IL-18. Conversely, silencing CLIC1 reduced these factors, validating its role in regulating apoptosis and pyroptosis. Mechanistically, we demonstrated that CLIC1 acts as a negative regulator of the Nrf2/HO-1 signalling pathway, a critical antioxidant defence mechanism. By inhibiting Nrf2/HO-1, CLIC1 amplified redox imbalance, oxidative stress, apoptosis, and pyroptosis. Furthermore, silencing CLIC1 enhanced Nrf2/HO-1 activity, alleviating neuronal injury. These findings propose CLIC1 as a novel molecular target for mitigating oxidative stress-induced neuronal damage and the inflammatory response in CIRI.

CLIC1 is a metamorphic protein that exists in both soluble cytoplasmic and membrane-associated forms, with the latter functioning as a chloride-selective ion channel [24]. It plays a key role in physiological processes such as stabilizing membrane potential, regulating pH, cell proliferation, fluid secretion, and maintaining cell volume [25]. Previous research has reported elevated CLIC1 levels in various neurological disorders and central nervous system cancers,

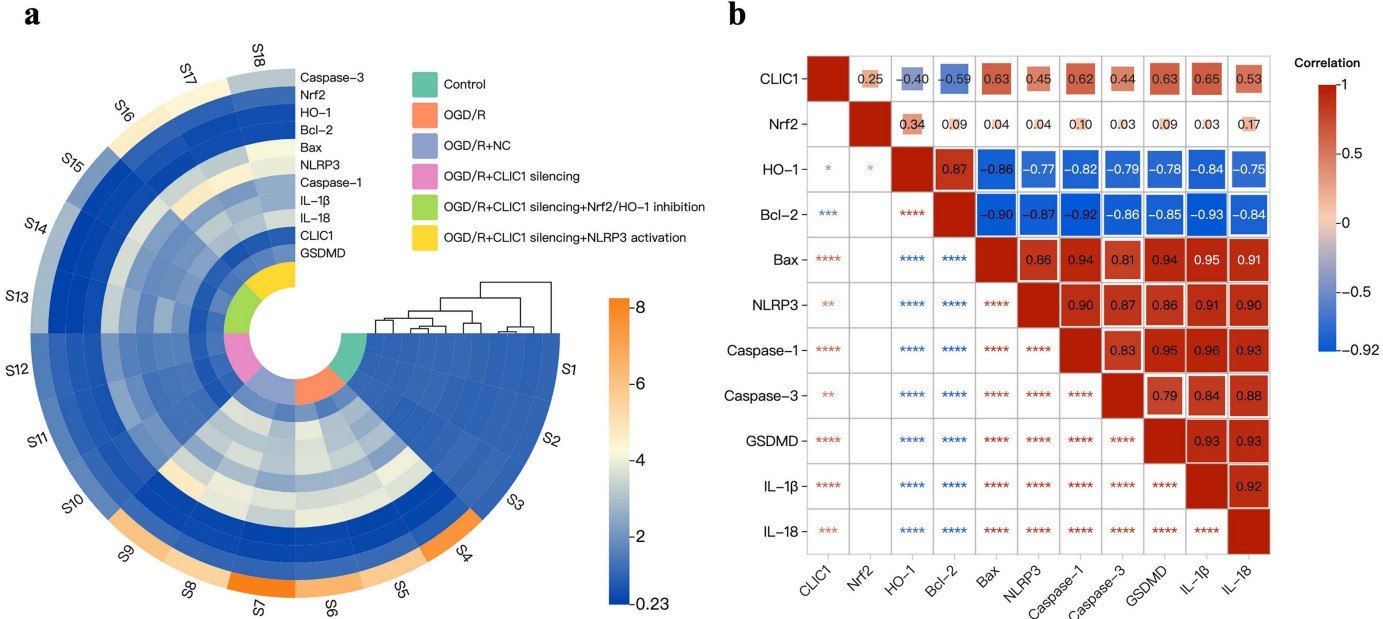

**Fig 8. Heatmaps of gene expression obtained from real-time PCR and the Spearman correlation analysis.** (a) Gene expression heatmap illustrates the relative expression levels of key genes (detected by real-time PCR) across all the six groups [the control, oxygen and glucose deprivation/reoxygenation (OGD/R), OGD/R+ negative control (NC), OGD/R+CLIC1 silencing, OGD/R+CLIC1 silencing+Nrf2/HO-1 inhibition, and OGD/R+CLIC1 silencing+NLRP3 activation]. Each row represents a specific gene, and each column represents a sample. (b) Correlation heatmap showing the Spearman correlation coefficients among the genes detected by real-time PCR. Each cell represents the correlation between two genes, with the colour gradient indicating the strength and direction of the correlation. *$P < 0.05$; **$P < 0.01$; ***$P < 0.001$; ****$P < 0.0001$.

such as Alzheimer's disease and stroke [26,27]. For example, CLIC1 upregulates in CD8 T cells and natural killer cells in atrial fibrillation thrombi, indicating its potential involvement in cytotoxic activity and tissue injury, thereby posing a risk of cardioembolic stroke [27]. During reperfusion, CLIC1 may be upregulated as part of the cellular defence mechanism to counteract oxidative damage and regulate intracellular chloride levels. However, the overexpression of CLIC1 disrupts redox homeostasis by promoting ROS generation and altering mitochondrial function. By interacting with specific proteins, CLIC1 was reported to promote mitochondrial fragmentation, increases membrane potential and ROS production, and induces a metabolic shift toward glycolysis in endothelial cells [28]. Whereas, inhibition of CLIC1 reduced oxidative stress-induced damage by lowering ROS levels, decreasing intercellular adhesion molecule 1 (ICAM1) and vascular cell adhesion molecule 1 (VCAM1), and enhancing the activity of antioxidant enzymes, including SOD [17]. Furthermore, we observed that CLIC1 regulated apoptotic markers, such as the pro-apoptotic Bax and anti-apoptotic Bcl-2, highlighting its role in apoptosis process in OGD/R-treated HT22 cells. Oxidative stress-induced apoptosis in CIRI can be alleviated via modulating the Bcl-2/Bax signalling pathway, which involves the downregulation of Bax and upregulation of Bcl-2 [29,30]. These findings suggest that CLIC1 may contribute to CIRI by fostering a redox imbalance and promoting apoptosis.

Concurrently, we found that CLIC1 upregulation promoted NLRP3-mediated pyroptosis in OGD/R-treated HT22 cells. CLIC1 has been reported to translocate to the plasma membrane upon cellular activation, where it modulates ion homeostasis and cell signaling, including the production and release of pro-inflammatory cytokines. A previous study [18] demonstrated that both CLIC1 and CLIC4 participate in regulating the NLRP3 inflammasome. And knocking down CLIC1 and CLIC4 impairs IL-1β transcription, apoptosis-associated speck-like protein containing a CARD (ASC) speck formation and mature IL-1 secretion. Furthermore, Carlini et al.[31] revealed that CLIC1 levels were significantly elevated in peripheral blood mononuclear cells during chronic central nervous system inflammation, suggesting its potential involvement in

inflammatory processes and acting as a potential marker of neurodegenerative processes. The NLRP3 inflammasome is pivotal in the inflammatory response associated with CIRI, triggering pyroptosis—a highly inflammatory form of programmed cell death [32]. By activating caspase-1 and GSDMD, NLRP3 facilitates the release of pro-inflammatory cytokine and release mature IL-1β and IL-18 by forming membrane pores [32–34], amplifying the inflammatory damage and leading to pyroptosis [23]. Additionally, NLRP3 inflammasomes can further worsen stoke or CIRI by compromising the integrity of the blood-brain barrier [7,35]. By inhibiting or blocking NLRP3 activation early, it may be possible to prevent CIRI by reducing inflammation of brain endothelial cells and stabilizing the blood-brain barrier [7]. Besides, ion homeostasis disruption upon reperfusion can lead to cellular swelling and damage. During cytoplasmic oxidation or pH changes, soluble CLIC proteins translocate to the cell membrane, functioning as anion channels to stabilize cation fluctuations [18,36]. These interconnected roles of NLRP3 inflammasome activation and CLIC1 regulation highlight the broader implications of inflammatory and ionic homeostasis pathways in CIRI. Therapeutic interventions targeting these mechanisms could provide multifaceted protection, curbing inflammation and stabilizing cellular environments, suggesting potential avenues for more effective treatment strategies in ischemic stroke management.

CLIC1 functions as an oxidative stress sensor and has been reported to regulate cellular senescence and function via the Nrf2/HO-1 pathway [17]. However, the precise initial downstream mechanisms remain unclear. Evidence suggests that CLIC1 inhibition increases intracellular $Ca^{2+}$ via the L-type Ca2+ channel (LTCC) [37]. In response to stressful oxidation conditions, CLIC1 modulates Nrf2 signaling by altering $Ca^{2+}$ homeostasis [38]. Besides, CLIC1 overexpression inhibits Nrf2 nuclear translocation and contributes to hydrogen peroxide-induced mitochondrial dysfunction and fission. As a key antioxidant transcription factor, Nrf2 also plays a significant role in reducing inflammation [39]. The overexpressed CLIC1 was found to supress Nrf2 nuclear translocation and Nrf2/HO-1 pathway [17]. HO-1 plays a critical role in breaking down heme into metabolites, which confers protection against oxidative injury and inflammation and resists to oxidative damage by mitigating cellular stress. Elevated HO-1 levels are widely acknowledged as protective, reducing ROS production, preventing neuronal apoptosis, and stabilizing cell membranes [40,41]. Importantly, activation of the Nrf2/HO-1 pathway has also been shown to suppress pyroptosis mediated by the NLRP3 inflammasome, caspase-1, and GSDMD, further underscoring its protective effects [42–45]. Conversely, suppression of the Nrf2/HO-1 pathway exacerbates NLRP3 activation and oxidative damage in OGD/R conditions, as demonstrated in studies linking Nrf2 downregulation with heightened inflammasome activity [46,47]. We aimed to clarify the mechanistic relationship between CLIC1 and NLRP3 in OGD/R-induced injury by showing that re-activation of NLRP3 in CLIC1-silenced cells restores pyroptosis, supporting NLRP3 as the key downstream effector of CLIC1. However, since OGD/R alone activates NLRP3, silencing NLRP3 directly may have been a more definitive approach, and we will refine this aspect in future studies.

Although apoptosis is often described as a non-inflammatory form of cell death, it has the capacity to trigger mild inflammatory responses [48]. We found that NLRP3-driven pyroptosis can in turn amplify oxidative stress and apoptotic processes, by regulating oxidative stress–related indices, MFI value of ROS and apoptosis rates. Together, these cell death mechanisms play distinct yet overlapping roles in both eliminating damaged cells and activating immune responses to CIRI. Apoptosis aids in clearing infected cells by phagocytes through efferocytosis, whereas pyroptosis, driven by the inflammasome, facilitates the removal of intracellular pathogens and the activation of immune responses [48,49]. Notably, our experiments measure signals of cell cluster, and the apparent co-activation of both death pathways may reflect different subpopulations of cells—some undergoing apoptosis, others pyroptosis. The correlation of CLIC1 with markers of both processes further indicates its central role in creating the stress environment, rather than the simultaneous occurrence of both forms of death in all cells. Additionally, recent research has demonstrated the possibility of transition and crosstalk between pyroptosis and apoptosis mechanisms. For example, Yuan et al. [50] found that the saponin monomer 13 of dwarf lilyturf tuber (DT-13) inhibits methylglyoxal-induced pyroptosis and facilitates a shift toward apoptosis by specifically modulating Caspase3/gasdermin E pathway in human umbilical vein endothelial cells. This underscores the plasticity between the two kinds of cell death form. Cao et al. [51] demonstrated that under diabetic and high glucose conditions, unfolded protein response pathways, the NLRP3 inflammasome, and apoptosis are activated in podocytes. NLRP3

overexpression promoted both high glucose-induced pyroptosis and apoptosis, and thioredoxin-interacting protein serves as a key link that mediated the mutual transformation between these two forms of cell death. However, apoptosis and pyroptosis relative contributions in CIRI depend on variables such as the specific experimental model used, the severity and duration of ischemia, and the extent of reperfusion injury. Considering this complexity, therapeutic strategies that integrate both anti-inflammatory and antioxidant or transformation of cell death forms hold great potential for managing CIRI.

Currently, the primary long-term objective of CIRI therapy aims to restore blood flow, reduce secondary damage, and minimize stroke-related neuronal cell death, disability, and mortality [52]. The treatment strategies for CIRI mainly include pre-ischemic preconditioning, post-ischemic preconditioning, and pharmacological preconditioning. Various pharmacological and interventional therapies have been developed to mitigate CIRI. However, the complexity of CIRI and the intricate interplay among various signalling pathways present significant challenges, leading to a lack of consensus in existing research [53,54]. Antioxidant and anti-inflammatory agents have shown promising neuroprotective measures in preclinical CIRI models [55]. Although there are no specific studies directly focusing on CLIC1 inhibitors for CIRI, previous studies [25,27,56] have suggested CLIC1 as a potential therapeutic target for central nervous system diseases, such as neurodegenerative diseases and ischemic stroke. For example, inhibition of CLIC1 channels may effectively suppress microglial proliferation and mitigated the neurotoxicity induced by Aβ-treated microglial cells [25]. Furthermore, a recent study [56] elucidates the regulatory role of circAPP in Alzheimer's disease via the miR-1906/CLIC1 axis during microglial polarisation. All these findings suggest that CLIC1 is a critical therapeutic target for central nervous system disorders and holds promising potential for treating CIRI. Furthermore, therapeutic regimens should aim to control CLIC1 temporally or spatially, adapting treatment to the disease stage or severity. The goal of targeted therapy is to inhibit its deleterious effects without completely abolishing its physiological function.

While our study primarily focuses on the mechanistic association between CLIC1 and Nrf2/HO-1 in regulating apoptosis and pyroptosis, additional oxidative stress-response pathways or regulatory networks, such as Wnt, nuclear factor kappa-B (NF-κB) and mitogen-activated protein kinase (MAPK) signalling [53,55,57], might also play a role in the observed phenomena. The activation of the canonical Wnt pathway during ischemic and reperfusion phases exerts organ-protective effects, whereas the activation of non-canonical Wnt pathway contributes to injury [53]. The suppression of NF-κB and MAPK signalling pathways was reported to inhibit the activation of microglia and reduce the production of inflammatory cytokines after OGD/R treatment, thereby protecting against neuronal injury [57]. These pathways and innovative approaches to treatment are critical contributors to cellular stress responses, inflammation, and apoptosis in CIRI. However, their potential crosstalk with CLIC1 remains to be fully understood. Further research is essential to develop and validate selective CLIC1 inhibitors to improve their clinical application. Moving forward, we hope to investigate CIRI treatment through molecular docking or macromolecular channel screening drugs.

## Conclusions

The critical role of CLIC1 in mediating oxidative stress, inflammation, and neuronal damage highlights its potential as a therapeutic target through regulation of the Nrf2/HO-1 signalling pathway. This targeted strategy for CIRI may address the current treatment gaps and substantially advance ischemic stroke therapy.

## Supporting information

**S1 File. Original PCR data for CLIC1, Nrf2, HO-1, Bcl-2, Bax, Caspase 3, NLRP3, Caspase 1, IL-1β, IL-18, and GSDMD in the control, OGD/R, NC, CLIC1 silencing, OGD/R+NC, and OGD/R+CLIC1 silencing groups.**
(XLS)

**S2 File. Original PCR data for CLIC1, Nrf2, HO-1, Bcl-2, Bax, Caspase 3, NLRP3, Caspase 1, IL-1β, IL-18, and GSDMD in the control, OGD/R, OGD/R+N, OGD/R+CLIC1 silencing, OGD/R+CLIC1 silencing+Nrf2/HO-1 inhibition, and OGD/R+CLIC1 silencing+NLRP3 activation groups.**
(XLS)

**S3 File. Original Western blot data for CLIC1, Nrf2, HO-1, Bcl-2, Bax, Caspase 3, NLRP3, Caspase 1, IL-1β, IL-18, and GSDMD in each group.**
(XLSX)

**S4 File. Original ELISA data for IL-1β and IL-18, as well as original SOD, CAT, GPX, and LDH data in each group.**
(XLSX)

**S5 File. Original flow cytometry data (ROS and apoptosis) and CCK-8 data in each group.**
(XLSX)

**S6 File. Original Western blot figures of each group.**
(PDF)

## Acknowledgments

We thank Charlesworth for scientific editing of this manuscript.

## Author contributions

**Conceptualization:** Chuang Sun.

**Data curation:** Honghai Chen, Chuanwen Yu, Yuhai Cao.

**Formal analysis:** Jingtong Xiong, Chuang Sun.

**Funding acquisition:** Jianlin Wu.

**Methodology:** Aodan Zhang.

**Supervision:** Xiaofeng Qu, Chuang Sun.

**Visualization:** Jianlin Wu.

**Writing – original draft:** Jingtong Xiong, Shuo Li.

**Writing – review & editing:** Aodan Zhang.

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
