## [Decision Letter · Decision Letter 0]

1 Dec 2024

Dear Dr. Sun,

Thank you for submitting your manuscript to PLOS ONE. After careful consideration, we feel that it has merit but does not fully meet PLOS ONE’s publication criteria as it currently stands. Therefore, we invite you to submit a revised version of the manuscript that addresses the points raised during the review process.

We look forward to receiving your revised manuscript.

Kind regards,

Songyun Zhao

Academic Editor

PLOS ONE

2. Thank you for stating the following financial disclosure: [This research was supported by the Liaoning Provincial Science and Technology Program (2022-MS-329), the General Program of National Natural Science Foundation of China�82071911�, and the Science and Technology Innovation Fund of Dalian�2021JJ12SN38�.]. Please state what role the funders took in the study. If the funders had no role, please state: "The funders had no role in study design, data collection and analysis, decision to publish, or preparation of the manuscript." If this statement is not correct you must amend it as needed. Please include this amended Role of Funder statement in your cover letter; we will change the online submission form on your behalf.

3. We note that your Data Availability Statement is currently as follows: [All relevant data are within the manuscript.] Please confirm at this time whether or not your submission contains all raw data required to replicate the results of your study. Authors must share the “minimal data set” for their submission. PLOS defines the minimal data set to consist of the data required to replicate all study findings reported in the article, as well as related metadata and methods (https://journals.plos.org/plosone/s/data-availability#loc-minimal-data-set-definition). For example, authors should submit the following data: - The values behind the means, standard deviations and other measures reported; - The values used to build graphs; - The points extracted from images for analysis. Authors do not need to submit their entire data set if only a portion of the data was used in the reported study. If your submission does not contain these data, please either upload them as Supporting Information files or deposit them to a stable, public repository and provide us with the relevant URLs, DOIs, or accession numbers. For a list of recommended repositories, please see https://journals.plos.org/plosone/s/recommended-repositories. If there are ethical or legal restrictions on sharing a de-identified data set, please explain them in detail (e.g., data contain potentially sensitive information, data are owned by a third-party organization, etc.) and who has imposed them (e.g., an ethics committee). Please also provide contact information for a data access committee, ethics committee, or other institutional body to which data requests may be sent. If data are owned by a third party, please indicate how others may request data access.

5. Please include a caption for figure 8.

6. Please ensure that you refer to Figure 7 in your text as, if accepted, production will need this reference to link the reader to the figure.

Additional Editor Comments (if provided):

Reviewers' comments:

Reviewer's Responses to Questions

**Comments to the Author**

1. Is the manuscript technically sound, and do the data support the conclusions?

Reviewer #1: Partly

Reviewer #2: Yes

2. Has the statistical analysis been performed appropriately and rigorously?

Reviewer #1: No

Reviewer #2: Yes

3. Have the authors made all data underlying the findings in their manuscript fully available?

Reviewer #1: No

Reviewer #2: Yes

4. Is the manuscript presented in an intelligible fashion and written in standard English?

Reviewer #1: No

Reviewer #2: Yes

Reviewer #1: This study examines the role of chloride intracellular channel 1 (CLIC1) in oxidative stress and inflammation during cerebral ischemia–reperfusion injury (CIRI) using an oxygen and glucose deprivation/reoxygenation (OGD/R) model in HT22 hippocampal neurons. The results show that CLIC1 is significantly upregulated under OGD/R conditions, leading to neuronal damage by suppressing the Nrf2/HO-1 signaling pathway. This suppression exacerbates oxidative stress, inducing apoptosis and NLRP3 inflammasome-mediated pyroptosis. Silencing CLIC1 alleviates these effects, reducing oxidative stress markers, inflammasome activity, and cell death. The findings highlight CLIC1's dual role in apoptosis and pyroptosis and suggest its potential as a therapeutic target to mitigate CIRI-induced neuronal injury by modulating the Nrf2/HO-1 pathway. I recommend publishing this manuscript with the inclusion of the following significant corrections, elaborated below, to further enhance its clarity, robustness, and scientific rigor.

1. Figures: Figures need higher resolution (300–600 dpi, per PLOS ONE guidelines) and more detailed, self-explanatory legends to improve clarity and interpretability.

2. Heatmaps: Gene expression and correlation heatmaps require better labeling for easier understanding.

3. Abbreviation Use: Abbreviations (e.g., NLRP3, GSDMD) should be defined at first mention and used consistently throughout the text.

4. Reference Formatting: The reference list does not fully adhere to journal guidelines. Full journal names (e.g., "Biomedicine & Pharmacotherapy" instead of "Biomed Pharmacother") and consistent DOI formatting are required.

5. Language and Grammar: Minor grammatical issues (e.g., “to promoted apoptosis” instead of “to promote apoptosis”) and wordiness reduce readability and need correction.

6. Novelty: The study needs to better emphasize its novel contribution, particularly the regulatory role of CLIC1 in the Nrf2/HO-1 pathway and its implications.

7. Mechanistic Clarity: While the study links CLIC1 to pyroptosis/apoptosis via Nrf2/HO-1, it does not explore alternative pathways or contributing factors, such as other stress-response mechanisms.

8. Experimental Design:

- Control Groups: Details on control groups, including sham-treated groups, are insufficient, raising concerns about baseline comparisons.

- Sample Size: With only three replicates per group, conclusions may lack statistical robustness. A power analysis is needed.

9. Pathway Validation: The evidence for Nrf2/HO-1 inhibition or silencing requires stronger validation, such as co-immunoprecipitation or experiments using pathway-specific inhibitors.

10. Methods: Experimental details, including validation of viral silencing and OGD/R treatment conditions, are inadequately described, which may hinder reproducibility.

Addressing these issues will significantly improve the manuscript’s clarity, scientific rigor, and alignment with journal guidelines.

Reviewer #2: In this manuscript, the authors use a well-established model (OGD/R-treated HT22 cells) and robust methodologies, including real-time PCR, western blotting, flow cytometry, and ELISA, to investigate apoptosis and pyroptosis. The study links CLIC1 overexpression with oxidative stress-induced apoptosis and NLRP3-mediated pyroptosis, providing mechanistic evidence via modulation of the Nrf2/HO-1 pathway. To enhance the study's rigor, several improvements could be made:

1. Lack of In Vivo Validation: While the cell model findings are significant, they may not fully translate to in vivo systems. Including animal studies would strengthen the conclusions. Please add in vivo experimental verification if conditions permit.

2. Some sentences are overly complex, leading to occasional ambiguity (e.g., in the introduction and discussion).

3. Several grammatical errors and awkward sentence structures are present. For example: "And this process led to an exacerbation of apoptosis..." → Avoid starting sentences with "And." "Upon CLIC1 silencing, all the mentioned indicators above exhibited an inverse trend..." → Simplify to: "Silencing CLIC1 reversed these effects." Repetition: The phrase "CLIC1 silencing" is overused. Vary terminology to improve readability.

4. In the discussion section, while the discussion highlights the importance of the findings, it occasionally reiterates results rather than expanding on their broader implications.

**Do you want your identity to be public for this peer review?** For information about this choice, including consent withdrawal, please see our Privacy Policy

Reviewer #1: No

Reviewer #2: No

---

## [Author Response · Author response to Decision Letter 1]

31 Jan 2025

Dear Editor and Reviewer

We would like to express our sincere appreciation to the editor for overseeing the peer review process and for providing us with the valuable opportunity to revise our manuscript. We are deeply grateful for your time, effort, and guidance, which have been invaluable in improving the quality and clarity of our work.

We extend our heartfelt thanks to the reviewers for their constructive and insightful comments, which have greatly contributed to enhancing the manuscript and the impact of our research.

In response to all the suggestions, we have carefully addressed each comment and revised the manuscript accordingly. We hope that the updated version meets the journal's publication requirements.

Please check the file of Response to reviewers.

Besides, due to the reviewer's request to increase the image resolution, we have re-uploaded the figures with a larger file size. We would be grateful for your understanding.

Thank you once again for your support and guidance throughout this process.Please do not hesitate to contact us if further revisions or additional steps are required.

Sincerely,

Chuang Sun

---

## [Decision Letter · Decision Letter 1]

21 Feb 2025

Dear Dr. Sun,

Thank you for submitting your manuscript to PLOS ONE. After careful consideration, we feel that it has merit but does not fully meet PLOS ONE’s publication criteria as it currently stands. Therefore, we invite you to submit a revised version of the manuscript that addresses the points raised during the review process.

We look forward to receiving your revised manuscript.

Kind regards,

Songyun Zhao

Academic Editor

PLOS ONE

**Journal Requirements:**

Reviewers' comments:

Reviewer's Responses to Questions

**Comments to the Author**

Reviewer #2: All comments have been addressed

2. Is the manuscript technically sound, and do the data support the conclusions?

Reviewer #2: Yes

3. Has the statistical analysis been performed appropriately and rigorously?

Reviewer #2: Yes

4. Have the authors made all data underlying the findings in their manuscript fully available?

Reviewer #2: Yes

5. Is the manuscript presented in an intelligible fashion and written in standard English?

Reviewer #2: Yes

**Reviewer #2: ** The author have addressed my concerns properly.

However, some of the references in the article are dated. It is recommended to add recently published studies on related topics in the field.

Please add the following references

1. Zhao, S., Zhuang, H., Ji, W. et al. Identification of Disulfidptosis-Related Genes in Ischemic Stroke by Combining Single-Cell Sequencing, Machine Learning Algorithms, and In Vitro Experiments. Neuromol Med 26, 39 (2024). https://doi.org/10.1007/s12017-024-08804-2.

2. Qian, Y., Yang, L., Chen, J. et al. SRGN amplifies microglia-mediated neuroinflammation and exacerbates ischemic brain injury. J Neuroinflammation 21, 35 (2024). https://doi.org/10.1186/s12974-024-03026-6.

3. Zhuang H, Lei W, Wu Q, Zhao S, Zhao Y, Zhang S, Zhao N, Sun J, Liu Y. Overexpressed CD73 attenuates GSDMD-mediated astrocyte pyroptosis induced by cerebral ischemia-reperfusion injury through the A2B/NF-κB pathway. Exp Neurol. 2025 Jan 18;386:115152. doi: 10.1016/j.expneurol.2025.115152.

**Do you want your identity to be public for this peer review?** For information about this choice, including consent withdrawal, please see our Privacy Policy

Reviewer #2: No

---

## [Author Response · Author response to Decision Letter 2]

19 Apr 2025

Dear Editor,

Thank you for providing us with the opportunity to revise our manuscript and for the valuable feedback from the reviewers.

We sincerely apologize for the oversight in updating our references and for not paying close attention to the status of cited literature during the long period of writing the article. This has been an important learning experience, and we will ensure greater diligence in future writing and reference management. We are grateful for your constructive guidance, which has benefited us greatly.

We have carefully reviewed each reference and made necessary adjustments. The added references have been incorporated into the revised manuscript.

We hope these revisions improve the scientific rigor and clarity of the manuscript. Should you require any further clarification or changes, please feel free to contact us.

Pleases check the file of Response to Reviewers.Thank you again for your time, support, and guidance.

Sincerely,

Chuang Sun

---

## [Decision Letter · Decision Letter 2]

12 Jun 2025

CLIC 1 down-regulates Nrf2/HO-1 signalling pathway promoting the apoptosis and pyroptosis in OGD/R-treated HT22 cells

PLOS ONE

Dear Dr. Sun,

Thank you for submitting your manuscript to PLOS ONE. After careful consideration, we feel that it has merit but does not fully meet PLOS ONE’s publication criteria as it currently stands. Therefore, we invite you to submit a revised version of the manuscript that addresses all the points raised during the review process.

We look forward to receiving your revised manuscript.

Kind regards,

Mária A. Deli, M.D., Ph.D.

Academic Editor

PLOS ONE

Additional Editor Comments:

Unfortunately none of the original 2 reviewers were available for the re-review of the manuscript. A third independent reviewer was asked to evaluate the second revision, who very carefully and critically read the paper and suggested several improvements. While no further experiments are needed, many small corrections, explanations need to be done. Importantly, the text also needs English editing!

Comments from PLOS Editorial Office: We note that one or more reviewers has recommended that you cite specific previously published works in an earlier round of revision. As always, we recommend that you please review and evaluate the requested works to determine whether they are relevant and should be cited. It is not a requirement to cite these works and you may remove them before the manuscript proceeds to publication. We appreciate your attention to this request.

Reviewers' comments:

Reviewer's Responses to Questions

**Comments to the Author**

Reviewer #3: (No Response)

2. Is the manuscript technically sound, and do the data support the conclusions?

Reviewer #3: Partly

3. Has the statistical analysis been performed appropriately and rigorously?

Reviewer #3: Yes

4. Have the authors made all data underlying the findings in their manuscript fully available?

Reviewer #3: No

5. Is the manuscript presented in an intelligible fashion and written in standard English?

Reviewer #3: No

Reviewer #3: This study demonstrates that manipulating the expression of CLIC1 may affect the development of inflammation and oxidative stress in an in vitro model of cerebral ischemia-reperfusion injury. However there are a few major gaps in the conclusions of this study.

1) How do the authors reconcile the fact that manipulating the same molecule CLIC1 may lead to two distinct cell death processes – apoptosis and pyroptosis that can not occur at the same time. Is there a preferential pathway to one or the other, or a state of the cell or a time window?

2) What is the rationale for the CLIC1 multiple roles in inflammation and oxidative stress in ischemic damage. Using the term “cellular adaptor“ (Discussion- line 5) will not do.

3) It is never disclosed what is the basis of the hypothesis that CLIC1 would suppress Nrf2/HO-1. In fact, it is not clearly disclosed what is the initial downstream pathway for the many CLIC1 effects.Only in Fig 1 “ARE“ is indicated without further mentioning or even explaining the abbreviation.

4) How do the authors reconcile their proposal of therapy by supressing CLIC1 and apoptosis/pyroptosis with their stating that CLIC1 has a cellular defensive mechanism (p. 16, 2pgph line 2-3) or that it stabilizes cation fluctuations during stress conditions and cytoplasmic oxidation (p17 last line, p18 lines 1-2) as well as with the sentence “Apoptosis aids in clearing infected cells, whereas pyroptosis facilitates the removal of intracellular pathogens and the activation of immune responses.“

In addition:

The experiment in Fig. 4 using activated NLRP3 after silencing CLIC1 to determine if NLRP3 mediates pyroptosis in OGD/R seems forced and superfluous since in the same Figure it was shown that NLRP3 is activated by OGD/R itself followed by pyroptosis markers. It is also merely strange to state in the same Fig legend (line 4-5) that activation of NLRP3 upregulates itself!

Page 4, 2nd pgph, line 5 : “CLIC1 ...removing excessive negative charge via anion channel“ However CLIC1 IS an anion channel

Legend to Fig. 1 is missing

First pgph on page 14 should belong to the part of text with Figure 4 .

There are a lot of typos and English is below acceptance level (eg “A recent study discovered that in umbilical vein endothelial cells, CLIC1 high-expressed and impaired the ability of vascular cells to resist oxidative stress to speed up cellular senescence, whose inhibition conversely perform a protective effect on preventing cellular aging and disorder“)

The authors claim that all relevant data are within the ms but this is not the case – jus the means and errors can be found.

Why do the authors disclose ethical review of animal welfare when they used no animals in their experiments? Only commercially available animal cell line was used

**Do you want your identity to be public for this peer review?** For information about this choice, including consent withdrawal, please see our Privacy Policy

Reviewer #3: No

---

## [Author Response · Author response to Decision Letter 3]

22 Jul 2025

Dear Editor and Reviewer,

We sincerely appreciate the time and effort you have dedicated to reviewing our manuscript. We are grateful for the rigorous, professional, and constructive feedback provided throughout the review process.

Our manuscript has undergone two rounds of revision—a major revision (PONE-D-24-49743_R1) and a minor revision (PONE-D-24-49743_R2)—prior to this submission. However, we noticed that the latest reviewer’s comments appear to be based on the original version of the manuscript (PONE-S-24-63293) rather than the most recently revised version.We have uploaded pdf files of the 3 previous versions of the manuscript in the attachment “others”, so that you can review them on demand.

Given that the previous reviewers and editor have already provided valuable suggestions to enhance the language, literature citations, readability, logical flow, and scientific rigor of the paper, we have made further refinements based on the latest revised version (PONE-D-24-49743_R2) rather than reverting to the original submission.

To facilitate your review, we have:

• Included a point-by-point response to all comments in file of Respond to Reviewers.

• Highlighted all new revisions in yellow in the revised manuscript.

We deeply value your expertise and the thoughtful critique, which has significantly strengthened our work. Should you have any additional questions or concerns, please do not hesitate to contact us.

Thank you once again for your time and consideration.

Best regards,

Chuang Sun

---

## [Decision Letter · Decision Letter 3]

5 Aug 2025

Dear Dr. Sun,

Thank you for submitting your manuscript to PLOS ONE. After careful consideration, we feel that it has merit but does not fully meet PLOS ONE’s publication criteria as it currently stands. Therefore, we invite you to submit a revised version of the manuscript that addresses the points raised during the review process.

We look forward to receiving your revised manuscript.

Kind regards,

Mária A. Deli, M.D., Ph.D.

Academic Editor

PLOS ONE

Journal Requirements:

Reviewers' comments:

Reviewer's Responses to Questions

**Comments to the Author**

Reviewer #3: All comments have been addressed

2. Is the manuscript technically sound, and do the data support the conclusions?

Reviewer #3: Yes

3. Has the statistical analysis been performed appropriately and rigorously?

Reviewer #3: Yes

4. Have the authors made all data underlying the findings in their manuscript fully available?

Reviewer #3: No

5. Is the manuscript presented in an intelligible fashion and written in standard English?

Reviewer #3: Yes

Reviewer #3: Comments are addressed thoroughly and successfully. Nevertheless, it would be more appropriate in the revised ms to use examples from CNS and not from cancer (Discussion Pgph 6) or vascular and bronchial tissues (Introduction pgph 3).

In addition, text in Duscussion pgph 2 5th line from the end should be changed to read "...highlighting its role in the apoptotic process..."

**Do you want your identity to be public for this peer review?** For information about this choice, including consent withdrawal, please see our Privacy Policy

Reviewer #3: **Yes: ** Pavle Andjus

---

## [Author Response · Author response to Decision Letter 4]

1 Sep 2025

Dear Editor and Reviewer,

We would like to express our sincere gratitude for your time, effort, and constructive feedback throughout the review process. Your insightful comments and valuable suggestions have been instrumental in helping us improve the quality and clarity of our manuscript.

To facilitate your review, we have:

• Provided a detailed, point-by-point response to all comments in this letter.

• Highlighted all relevant modifications to the main text in yellow.

We deeply appreciate your careful evaluation and support, which have contributed significantly to strengthening our work. Thank you again for your guidance and for the opportunity to revise and resubmit our article.

Please do not hesitate to contact us if you have any further questions or concerns.

Best regards,

Chuang Sun

---

## [Editor Report · Decision Letter 4]

3 Sep 2025

CLIC 1 down-regulates Nrf2/HO-1 signalling pathway promoting the apoptosis and pyroptosis in OGD/R-treated HT22 cells

PONE-D-24-49743R4

Dear Dr. Sun,

We’re pleased to inform you that your manuscript has been judged scientifically suitable for publication and will be formally accepted for publication once it meets all outstanding technical requirements.

Kind regards,

Mária A. Deli, M.D., Ph.D.

Academic Editor

PLOS ONE

Additional Editor Comments (optional):

I thank the authors to do the final round of amendments. Since the last reviewer already accepted the paper, no need for further rounds of review, I consider the manuscript scientifically acceptable for publishing.